# Investigation of superspreading COVID-19 outbreak events in meat and poultry processing plants in Germany: A cross-sectional study

Roman Pokora[1,2]*, Susan Kutschbach[1], Matthias Weigl[1], Detlef Braun[1], Annegret Epple[1], Eva Lorenz[2,3,4], Stefan Grund[1], Juergen Hecht[1], Helmut Hollich[1], Peter Rietschel[1], Frank Schneider[1], Roland Sohmen[1], Katherine Taylor[2], Isabel Dienstbuehl[1]

1 Division of Prevention, Berufsgenossenschaft Nahrungsmittel und Gastgewerbe (BGN), Germany, 2 Institute of Medical Biostatistics, Epidemiology and Informatics, University hospital of the Johannes Gutenberg-University Mainz, Mainz, Germany, 3 Infectious Disease Epidemiology, Bernhard Nocht Institute for Tropical Medicine, Hamburg, Germany, 4 German Center for Infection Research (DZIF), Hamburg-Borstel-Lübeck-Riems, Germany

* pokora@uni-mainz.de

**Data Availability Statement:** All relevant data are within the manuscript and the dataset is available under https://doi.org/10.5281/zenodo.4692642. The legal basis for the collection of the study data

## Abstract

Since May 2020, several COVID-19 outbreaks have occurred in the German meat industry despite various protective measures, and temperature and ventilation conditions were considered as possible high-risk factors. This cross-sectional study examined meat and poultry plants to assess possible risk factors. Companies completed a self-administered questionnaire on the work environment and protective measures taken to prevent SARS-CoV-2 infection. Multivariable logistic regression analysis adjusted for the possibility to distance at least 1.5 meters, break rules, and employment status was performed to identify risk factors associated with COVID-19 cases. Twenty-two meat and poultry plants with 19,072 employees participated. The prevalence of COVID-19 in the seven plants with more than 10 cases was 12.1% and was highest in the deboning and meat cutting area with 16.1%. A subsample analysis where information on maximal ventilation rate per employee was available revealed an association with the ventilation rate (adjusted odds ratio (AOR) 0.996, 95% CI 0.993–0.999). When including temperature as an interaction term in the working area, the association with the ventilation rate did not change. When room temperatures increased, the chance of testing positive for COVID-19 (AOR 0.90 95% CI 0.82–0.99) decreased, and the chance for testing positive for COVID-19 for the interaction term (AOR 1.001, 95% CI 1.000–1.003) increased. Employees who work where a minimum distance of less than 1.5 m between workers was the norm had a higher chance of testing positive (AOR 3.61; 95% CI 2.83–4.6). Our results further indicate that climate conditions and low outdoor air flow are factors that can promote the spread of SARS-CoV-2 aerosols. A possible requirement for pandemic mitigation strategies in industrial workplace settings is to increase the ventilation rate.

is section 1, section 9 paragraph 8, section 14 and 207 of the Social Code (SGB VII) and the statutes of the BGN. All persons involved in the project are subject to the obligation to maintain confidentiality in accordance with Section 35 SGB I. The analysis presents insurance data collected during the legal mandate with ongoing follow-up data collection. Original questionnaires and measurement reports are not made available for the scientific community outside the established and controlled workflows and algorithms. To meet the general idea of verification and reproducibility of scientific findings, we offer access to the original data at the local database in accordance with the data protection protocol upon request at any time. The study team, which constitutes a member of each involved department and the coordinating principal investigator of the study (Dr Pokora) decides on internal and external access of researchers and use of the data based on a research proposal to be supplied by the researcher. Interested researchers make their requests to the coordinating principal investigator of the study (roman.pokora@bgn.de).

**Funding:** The authors received no specific funding for this work.

**Competing interests:** The authors have declared that no competing interests exist.

## Introduction

Numerous COVID-19 outbreaks, the disease caused by the novel coronavirus (SARS-CoV-2), in several meat processing plants around the globe have been described [1–3]. In Germany alone, the media reported approximately 3,654 positive SARS-CoV-2 tests (in the following also denoted as COVID-19 cases) in meat processing plants, and public health authorities reported 2,819 positive cases tested among employees from meat processing plants to the Berufsgenossenschaft Nahrungsmittel und Gastgewerbe (BGN; English: German Social Accident Insurance Institution for the foodstuff and catering industry; current as of 28.8.2020).

On 16 April 2020, the German Federal Ministry of Labor and Social Affairs issued a recommendation for working during the pandemic. This SARS-CoV-2 occupational safety standard made clear COVID-19 prevention recommendations on distance, face coverings, personal protective equipment, cleaning, and hand hygiene measures. However, there have still been at least 16 outbreaks so far in which at least 10 workers per meat processing plant tested positive in a meat processing company. The larger outbreaks had implications for entire communities, as interventions were implemented to contain the spread [4]. At first, employee accommodation (temporary housing for these primarily temporary and contract workers) in the meat industry were suspected as a possible distribution factor for SARS-CoV-2 [5] and seem to correlate positively with the infection figures [6]. However, living together and going to work together does not seem to be the sole explanation, which is why working conditions have increasingly been considered as a relevant risk factor.

Slaughterhouses and meat processing plants are characterized by particular working conditions. For food safety reasons some areas maintain a low ambient temperature [7] (REGULATION (EC) No 853/2004), which may result in better survival of SARS-CoV-2 at lower temperatures compared to room temperature [8–10]. Results from cross-sectional studies sometimes imply an association between COVID-19 infection and temperature and humidity [11–13], but one study did not find an association [14]. In addition, work areas requiring strenuous manual labor mean employees have an increased breathing rate and volume, and noisy work areas necessitate loud speech for communication [7]. As a result, an employee infected with SARS-CoV-2 will produce numerous SARS-CoV-2 aerosols [15–17]. Once released into the air, they remain in the environment for different durations depending on the ventilation, which provides fresh air and dilutes the concentration of airborne contagions [17–19]. The extent of this air exchange appears to affect the distribution of aerosols containing SARS-CoV-2 [17,19], which are suspected of being the main vector for COVID-19 infections in the meat industry [7]. The air exchange in rooms with a ventilation system is realized by moving outdoor air inside the building with the possible addition of circulating air flow. Some meat industry tasks require working with little distance between employees for entire shifts, for example in the meat processing and in the packaging area [5].

The dynamics of SARS-CoV-2 are not yet well understood. First studies have observed more infections in meat industry areas where distances between employees of less than two meters is the norm [20], but distances of up to 8 meters could also play a role [6]. In view of this research gap, we want to explore the impact of these particular working conditions on SARS-CoV-2 transmission. From the previous literature, we therefore derived the following hypotheses:

1. Higher infection rates in employees are observed in work areas with low temperatures.

2. In work areas with lower air flow (fresh air supply) per employee, higher infection rates in employees are observed more often than in work areas with higher air flow per employee.

3. In work areas where a minimum distance of 1.5 meters cannot be maintained, higher infection rates in employees are observed than in work areas meeting this minimum distance requirement.

4. The association of the outdoor air flow per employee is amplified in cooled work areas, which is why we expect a multiplicative interaction effect.

## Methods

### Setting, participants, and study design

In a cross-sectional study, twenty-six companies were contacted by BGN Prevention Management with a cover letter containing information on the study and a request to complete a questionnaire about protective measures against new infections caused by SARS-CoV-2. 17 companies had known infections and 9 had none in the period from the end of June to the beginning of September 2020.

Previously designated contacts at each site received the questionnaire per email and were asked to provide information about the employees, working conditions, and organization measures to prevent the spread of COVID-19 for the work areas in their establishments. They were asked to return the questionnaire within one week.

In addition to the questionnaire, the objectively measurable parameters on the ventilation technology in cooled areas of the plants with many infected employees are collected on-site. So far, this has already been done in two companies.

### Questionnaire

The questionnaire was designed based on the specific features of the slaughterhouses and the meat processing industry suspected of being related to infection occurrence (distance between employees, temperature and ventilation conditions) (see S1–S3 Appendices). In the meat industry, individual work areas within a company fundamentally differ from each other depending on the process stages to be carried out. The differentiation of the work areas follows the logical production sequence of animal processing, right up to the packaged food.

Information on separate work areas as well as the company in general was collected. These included questions about the number of employees, employment relationship types, the general housing and transport situation, especially for particularly at-risk workers (temporary and contract workers), the types of face coverings used, and other questions on infection protection within the framework of the current SARS-CoV-2 occupational safety standards (e.g. time shift, occupational health organization, preventive measurement promoting social distancing).

In the initial questionnaire (S1 Appendix), information about the extent of outdoor air and air circulation were collected. To be able to use the data on the ventilation individually in the analysis, data on the outdoor air flow were required, which is why a second version of the questionnaire was developed. Additionally, we were not able to distinguish case numbers between regular and temporary and contract workers and therefore added a new column in the questionnaire (see S2 and S3 Appendices). The definition and the exact difference of temporary and contract workers is not selective, and therefore we decided to combine the two groups in the analysis and did not collect more information for these two groups. After this decision, we again contacted the organizations and collected this additional information.

### Operationalization

Plants also provided information regarding specified interventions and prevention efforts that were implemented. The free text information was classified manually by two raters. If anything was unclear, this was discussed in a group of three people and a decision was made.

The number of temporary and contract workers was determined by subtracting the number of regular workers from the total number of employees. Similarly, the number of infected

temporary and contract workers was calculated by subtracting the number of infected regular workers from infected workers in total. The point prevalence is given as the percentage of workers who had reported a positive test for COVID-19 to their employer by a certain point in time, or as the number of diseased workers per 100 workers. The prevalence is calculated separately for all employees as well as for permanent or temporary staff. The combined prevalence for all companies is a period-based point prevalence and combines the different point prevalences in the plants.

In terms of infection, plants were classified into the categories of no infections (0 infected), a few infected employees (1–10 infected), and many infected employees (>10 infected). The type of breaks taken during work were categorized as: breaks taken freely, fixed breaks with time shift, and fixed breaks without time shift. Fixed breaks without time shift mean that all workers have their breaks at the same time. Fixed breaks with time shift mean that there is a break pattern where different working groups, departments or divisions have different time windows for their break.

The ventilation rate (hereinafter outdoor air flow (OAF)) on its own, is not sufficient for analysis. A high OAF value does not necessarily translate into a high amount of fresh air per employee. The absolute value is furthermore strongly influenced by the size of the room. Therefore, we made two assumptions in order to be able to include this value in the evaluations:

1. To evaluate the amount of outdoor air in $m^3/h$ as a measure to prevent the spread of SARS--CoV-2, the OAF is divided by the number of employees in a work area.

2. If the employees work in shifts in a work area, only the employees who work during the same shift are in this work area at a time, so we divide the number of employees in a work area by the number of shifts.

This resulted in the following formula for the OAF per employee in a work area:

$$Outdoor\ air\ flow\ per\ employee\ in\ a\ working\ area = \frac{Outdoor\ air\ flow\ in\ m3/h}{\frac{Number\ of\ employees\ in\ a\ working\ area}{Number\ of\ shifts\ in\ the\ working\ area}}$$

The room temperature was given in degrees Celsius and was also included in the evaluation as an interval scaled variable.

## Statistical analysis

Data entry, cleaning, and evaluation were performed using SAS 9.4 and IBM SPSS Statistics Version 25. Participants were included in the main analysis if they were working in a plant with at least 10 employees who had been infected with SARS-CoV-2 at the time of the questionnaire. Working characteristics of the study subjects were expressed by mean values for continuous variables and by relative and absolute frequencies for discrete variables. Beside the range of prevalences in the participating plants, results and information about individual companies are not presented.

Multivariable logistic regression analysis was performed to study the association between the reported COVID-19 infection of an employee as the outcome variable and the explaining working factors. Therefore, we report the crude odds ratios (OR) and an adjusted model including the following independent exposure variables: (1) comply with the minimum distance of 1.5 meters as a dummy variable; (2) room temperature in degrees Celsius; (3) having a ventilation system as a dummy variable; (4) type of work break as an ordinal variable and (5) type of contractual relationship with the company as a dummy variable. Additional analyses

were carried out using data for all employees for whom we had information on OAF. In a next step, we also excluded the first processing steps due to the use of carbon dioxide to stun some animals (e.g. pigs) in the slaughter area. These areas have a process related high ventilation rate. Therefore, the relationship between infection and OAF could be biased in these areas. In this final analysis, we changed variable (3) to OAF per employee in a work area and introduced a multiplicative interaction term for temperature and OAF per employee in a work area. During the on-site visits, it became apparent that air-conditioning had subsequently been installed in a meat processing area after the superspreading event. Therefore, we excluded workers from this work area in this plant from the ventilation sub-analysis.

Missing temperature information was replaced in the non-cooled work areas by the assumption of a seasonally slightly increased room temperature of 22˚C. All other missing values were included in tables of the descriptive analyses as an additional category for the respective variable (n, %).

## Results

Of the 22 participating companies, information was collected for 19,072 employees, including 880 infected workers (S1 Table). Infected workers screened before entering the plant are not reported. One plant had 356 infected workers, representing 40.5% of all infected workers in our study.

Our study includes seven plants with many infected workers, with a prevalence between 2.94 (95% CI 2.22–3.87) to 35.10 (95% CI 29.40–41.27) infections per 100 employees. A total of 856 infected individuals were counted in these plants. Five plants had fewer than 10 infected individuals, and ten plants (one plant took part although it was not contacted) had none. Plants in the low-infection category had an average period-based prevalence of 0.6%, and plants with many infected workers had an average period-based prevalence of 10.98%. The seven plants with many infected workers were in six cases slaughterhouses and/or meat processing facilities and one plant for meat and sausage production. Two out of the six slaughterhouses and meat processing facilities also had small divisions specialized for meat and sausage production.

Among the 22 facilities, information on interventions and prevention efforts was available for 20 (91%) facilities. Overall, 16 (72%) facilities reported a SARS-CoV-2 testing strategy, 11 (50%) planned to improve or already had improved ventilation, 10 (45%) installed physical barriers, and 6 (27%) required universal face covering (Table 1). Additional protective measures were reported separately in break rooms, canteen, changing rooms and at the entry of the facility.

The use of face coverings varied in the facilities depending on the working place (Table 2). Overall, more types of masks were named by companies without an infection in the workforce. In fifteen (68%) of 22 facilities, medical face coverings were used. FFP2 masks were mainly used in companies with zero infected people, while medical masks were already more common in all companies.

In total, 7,798 employees of the sample worked in the plants with many infected employees (Table 3). 949 (12.2%) workers had missing information on the distance variable. An additional 83 subjects working in cooled areas (1.1% (overall 562 subjects, 7.2%)) were excluded from the analysis because they had missing information on the temperature and 244 subjects (3.1% (overall 1044 subjects, 13.39%)) were excluded from the analysis because they had missing information on the type of work break, resulting in 6,522 employees eligible for the main analysis. Of these employees, we collected information on the air flow volume per employee for 2,786 employees (35.7%), who were eligible for sub-analysis. Nearly 73% of the study

**Table 1. Interventions and prevention efforts implemented by facilities in response to COVID-19 among workers or planned if there would be a case in 22 meat and poultry processing facilities—June–September 2020.**

| Intervention/Prevention effort | Facilities no. (%), who named the implementation in the questionnaire | | | |
| --- | --- | --- | --- | --- |
| | Many infected workers | Few infected workers | No infected workers | Overall |
| | N = 7 | N = 5 | N = 10 | N = 22 |
| SARS-CoV-2 testing strategy | 6 (85.71) | 4 (80) | 6 (60) | 16 (72.73) |
| Improved ventilation | 5 (71.43) | 5 (100) | 1 (10) | 11 (50) |
| Installed physical barriers between workers | 4 (57.14) | 3 (60) | 3 (30) | 10 (45.45) |
| Required universal face covering | 3 (42.86) | 2 (40) | 1 (10) | 6 (27.27) |
| Worker fever measure screening on entry | 1 (14.29) | 0 (0) | 2 (20) | 3 (13.64) |
| Introduced distance rules | 2 (28.57) | 0 (0) | 1 (10) | 3 (13.64) |
| Introduced quarantine rules | 1 (14.29) | 0 (0) | 2 (20) | 3 (13.64) |
| Staggered shifts | 1 (14.29) | 0 (0) | 1 (10) | 2 (9.09) |
| Changed cleaning and disinfection | 2 (28.57) | 0 (0) | 0 (0) | 2 (9.09) |
| Closure of production lines | 1 (14.29) | 1 (20) | 0 (0) | 2 (9.09) |
| Improve markings and signage | 1 (14.29) | 0 (0) | 1 (10) | 2 (9.09) |
| Controls of compliance with the rules | 1 (14.29) | 0 (0) | 1 (10) | 2 (9.09) |
| Other* | 3 (42.86) | 0 (0) | 3 (30) | 6 (27.27) |
| No mentioning | 0 (0) | 0 (0) | 2 (20) | 2 (9.09) |

*The category 'Other' includes restricting of stay capacity, home office, information about SARS-CoV-2, spatial equalization in the production lines, disinfection, training.

population was temporary or contract workers. All working characteristics for the study sample are presented in Table 1.

The temperatures across work areas differ in our sample due to the different hygienic requirements and processing steps (Tables 3 and 4). Deboning and cutting area, meat and sausage production, packaging, and commissioning must always be cooled. In our sample, these work areas have, on average, a significantly lower temperature (3.9°C—8.9°C) than all other work areas (13.5°C—22°C). Overall, 78.9% of the employees had to work in areas where the temperature is below 12°C and, for technical reasons, for 60.2% of the employees the minimum distance of at least 1.5 m cannot be guaranteed (Table 3). Most of these are working in the deboning and meat cutting area (Table 4). In addition to the type of cooling, it was also of interest to examine the extent to which the air conditioning systems are attached to an outdoor air supply. The values for OAF vary on average between 450 $m^3$/h in the workshop and up to 51,470 $m^3$/h in the delivery. If the values for OAF are converted to a person level, the delivery is 20,176 $m^3$/h, the anesthesia/slinging/hanging is 1,885 $m^3$/h, the slaughtering is 731 $m^3$/h, and the commissioning is 727 $m^3$/h, which lie clearly above the values of the other working areas.

**Table 2. Type of face covering by facilities in response to SARS-CoV-2 in 22 meat and poultry processing facilities—June–September 2020.**

| Type of face covering | Facilities no. (%), who named the implementation in the questionnaire | | | |
| --- | --- | --- | --- | --- |
| | Many infected workers | Few infected workers | No infected workers | Overall |
| | N = 7 | N = 5 | N = 10 | N = 22 |
| Medical face covering | 5 (71.43) | 3 (60) | 7 (70) | 15 (68.18) |
| Astronaut cap with face mask | 3 (42.86) | 4 (80) | 5 (50) | 12 (54.55) |
| FFP2 mask | 2 (28.57) | 1 (20) | 6 (60) | 9 (40.91) |
| Other | 2 (28.57) | 4 (80) | 7 (70) | 13 (59.09) |

**Table 3. COVID-19 among workers in meat processing plants by area of workplace in plants with many infected employees.**

| Characteristics | Companies with work area | Overall (N) | Without excluded (N,%) | Regular workers (n, %) | Temporary and contract workers (n,%) |
|---|---|---|---|---|---|
| Overall | 7 | 7798 | 6522 (100%) | 1769 (27.15%) | 4753 (72.85%) |
| **Work area** | | | | | |
| Delivery | 6 | 38 | 38 (100%) | 14 (36.84%) | 24 (63.16%) |
| Anesthesia/slinging/hanging | 6 | 124 | 124 (100%) | 48 (38.71%) | 76 (61.29%) |
| Slaughter | 6 | 524 | 524 (100%) | 60 (11.45%) | 464 (88.55%) |
| Deboning and meat cutting area | 7 | 3360 | 3360 (100%) | 644 (19.17%) | 2716 (80.83%) |
| Meat production | 3 | 484 | 290 (100%) | 110 (37.93%) | 180 (62.07%) |
| Sausage production | 3 | 143 | 138 (100%) | 55 (39.86%) | 83 (60.14%) |
| Smoking of meat | 1 | 4 | 4 (100%) | 1 (25%) | 3 (75%) |
| Packaging | 7 | 1212 | 1186 (100%) | 97 (8.18%) | 1089 (91.82%) |
| Commissioning/Loading | 6 | 255 | 163 (100%) | 104 (63.80%) | 59 (36.20%) |
| Garage | 7 | 241 | 188 (100%) | 149 (79.26%) | 39 (20.74%) |
| Cleaning of slaughter and production | 7 | 274 | 14 (100%) | 14 (100%) | 0 (0%) |
| Administration | 7 | 493 | 493 (100%) | 473 (95.94%) | 20 (4.06%) |
| Other work areas | 7 | 646 | - | - | - |
| **Temperature (Mean (SD))** | Missing | 10.49 (7.05; n = 7236) | 9.62 (6.46) | 12.85 (8.03) | 8.41 (5.28) |
| 2–6° C | 562 | 3287 | 3169 (100%) | 641 (20.23%) | 2528 (79.77%) |
| 7–12° C | | 1985 | 1980 (100%) | 381 (19.24%) | 1599 (80.76%) |
| >12° C | | 1964 | 1373 (100%) | 747 (54.41%) | 626 (45.59%) |
| **Social distance** | Missing | | | | |
| Minimum distance 1.5 meters | 949 | 2920 | 2593 (100%) | 823 (31.74%) | 1770 (68.26%) |
| Minimum distance less than 1.5 meters | | 3929 | 3929 (100%) | 946 (24.08%) | 2983 (75.92%) |
| **Lunch break** | | | | | |
| Break freely selectable | 1044 | 350 | 350 (100%) | 338 (96.57%) | 12 (3.43%) |
| Fixed breaks not shifted in time | | 2178 | 1999 (100%) | 676 (33.82%) | 1323 (66.18%) |
| Fixed breaks shifted in time | | 4226 | 4173 (100%) | 755 (18.09%) | 3418 (81.91%) |
| **Infected** | - | 856 | 789 (100%) | 148 (18.78%) | 641 (81.22%) |
| **Period-based prevalence (% (95% CI))** | | 10.98 (10.30–11.69) | 12.10 (11.33–12.91) | 8.37 (7.16–9.75) | 13.49 (12.54–14.49) |
| **Maximal outdoor air flow (OAF) N** | | 2786 | 2786 | 628 | 2158 |
| **Mean (SD)** | - | 20,318.79 (12561.01) | 20,318.79 (12561.01) | 10,689.25 (10,647.99) | 23,121.09 (11,657.80) |
| **OAF per employee (Mean (SD))** | - | 364.32 (2318.48) | 364.32 (2318.48) | 554.62 (4,166.75) | 308.94 (1,371.12) |

Abbreviations: SD, standard deviation; 95% CI, 95% confidence interval.

Table 5 shows the results of multivariable logistic regression analysis for having a positive COVID-19 test. Univariate analysis revealed a 71% greater chance of testing positive (OR 1.71; 95% CI 1.42–2.06) if the employee had a temporary contract rather than a permanent contract. But these differences disappeared when adjusted for the covariates. Across all models, employees who work where a minimum distance of less than 1.5 m between workers was the norm had a higher chance of testing positive (AOR 1.86; 95% CI 1.55–2.22). Having a ventilation system reduced the chance of testing positive and was lowest in the unadjusted model of the sub-group with only temporary and contract workers (OR 0.3; 95% CI 0.21–0.43). Workers in workplaces with higher room temperatures also had a lower chance of testing positive (AOR 0.98; 95% CI

**Table 4. Characteristics of the work areas in the plants with many infected employees.**

| Work area | Employees (N, %) | Infect-ed | Prevalence (% 95% CI)) | Temperat-ure (Mean (SD)) | Minimum distance less than 1.5 meters (n, %) | Lunch break (n, %) Break freely taken | Fixed breaks not shifted in time | Fixed breaks shifted in time | Maximal outdoor air flow (OAF) (Mean (SD, n)) | OAF per employee (Mean (SD, n)) |
|---|---|---|---|---|---|---|---|---|---|---|
| Delivery | 38 (0.58%) | 1 | 2.63 (0.47–13.49) | 20.68 (3.43) | 0 (0%) | 6 (1.71%) | 0 (0%) | 32 (0.77%) | 51,470.59 (9,836.91; 17) | 20,176.47 (17,462.23; 17) |
| ASH | 124 (1.9%) | 2 | 1.61 (0.44–5.69) | 21.53 (0.86) | 72 (1.83%) | 0 (0%) | 46 (2.3%) | 78 (1.87%) | 10,282.05 (13,007.39; 78) | 1,884.62 (5,879.75; 78) |
| Slaughter | 524 (8.03%) | 65 | 12.4 (9.85–15.50) | 20.51 (2.13) | 274 (6.97%) | 0 (0%) | 37 (1.85%) | 487 (11.67%) | 30,255.35 (14,805.57; 357) | 730.87 (995.68; 357) |
| Meat cutting area | 3,360 (51.52%) | 542 | 16.13 (14.93–17.41) | 6.51 (2.37) | 2,970 (75.59%) | 0 (0%) | 1435 (71.79%) | 1925 (46.13%) | 21,424.79 (11,481.36; 1,549) | 110.20 (59.43; 1,549) |
| Meat production | 290 (4.45%) | 26 | 8.97 (6.19–12.81) | 8.28 (2.92) | 110 (2.8%) | 0 (0%) | 0 (0%) | 290 (6.95%) | 3,500.0 (0.0; 110) | 31.82 (0.0; 110) |
| Sausage production | 138 (2.12%) | 6 | 4.35 (2.01–9.16) | 8.39 (0.21) | 0 (0%) | 0 (0%) | 0 (0%) | 138 (3.31%) | - | - |
| Smoking of meat | 4 (0.06%) | 0 | 0 (0.00–48.99) | 22.00 (0.00)* | 0 (0%) | 0 (0%) | 0 (0%) | 4 (0.1%) | - | - |
| Packaging | 1,186 (18.18%) | 95 | 8.01 (6.60–9.69) | 6.23 (1.71) | 370 (9.42%) | 0 (0%) | 378 (18.91%) | 808 (19.36%) | 19,279.48 (3,375.56; 458) | 94.98 (17.12; 458) |
| Commissioning | 163 (2.5%) | 23 | 14.11 (9.59–20.28) | 2.78 (1.86) | 0 (0%) | 0 (0%) | 55 (2.75%) | 108 (2.59%) | 20,000.0 (0.0; 55) | 727.27 (0.0; 55) |
| Garage | 188 (2.88%) | 14 | 7.45 (4.49–12.11) | 21.84 (1.26) | 0 (0%) | 3 (0.86%) | 14 (0.7%) | 171 (4.1%) | 450.0 (0.0; 40) | 34.62 (0.0; 40) |
| Cleaning | 14 (0.21%) | 1 | 7.14 (1.27–31.47) | 14.14 (1.41) | 0 (0%) | 0 (0%) | 4 (0.2%) | 10 (0.24%) | - | - |
| Administration | 493 (7.56%) | 14 | 2.84 (1.70–4.71) | 22.00 (0.00) | 133 (3.39%) | 341 (97.43%) | 30 (1.5%) | 122 (2.92%) | 5,000.0 (0.0; 122) | 40.98 (0.0; 122) |
| Overall | 6522 (100%) | 789 | 12.10 (11.33–12.91) | 9.62 (6.46) | 3929 (100%) | 350 (100%) | 1999 (100%) | 4173 (100%) | 20,318.79 (12,561.01; 2786) | 364.32 (2318.48; 2786) |

Abbreviations: SD, standard deviation; 95% CI, 95% confidence interval.

0.96–0.99). This relationship was greater among regular workers compared to temporary and contract workers. Workplaces where breaks could be taken as suited (AOR 0.37; 95% CI 0.20–0.70) and workplaces where fixed breaks were not shifted over time (AOR 0.22; 95% CI 0.17–0.28) had a lower risk of testing positive than workers with fixed breaks shifted over time.

The subsample analysis including information on maximal OAF per employee revealed an association for OAF with COVID-19 infections when the delivery, stunning/slinging/hanging, and slaughter areas were excluded from the analysis (AOR 0.996 95% CI 0.993–0.999) (Table 6). When including an interaction term for temperature and OAF, the chances of testing positive for CVOID-19 in higher-temperature rooms became lower (AOR 0.90 95% CI 0.82–0.99). The association of OAF with COVID-19 infections became also lower (AOR 0.984; 95% CI 0.971–0.996), and a 0.1% greater chance of testing positive for COIVID-19 developed for the interaction term (AOR 1.001 95% CI 1.000–1.003).

## Sub-analysis

During on-site visits, we learned that an air-conditioning installation had subsequently been installed in a meat processing area. Excluding the workers from this work area in this plant

**Table 5. Results of the logistic regression analysis for COVID-19 infections.**

| Characteristics | Overall N = 6,522 | | Regular workers n = 1,769 | | Temporary and contract workers n = 4,753 | |
|---|---|---|---|---|---|---|
| | OR (95% CI) | AOR (95% CI) | OR (95% CI) | AOR (95% CI) | OR (95% CI) | AOR (95% CI) |
| Minimum distance at least 1.5 meters | 1 | 1 | 1 | 1 | 1 | 1 |
| Minimum distance less than 1.5 meters | 1.789 (1.519–2.107) | 1.856 (1.552–2.221) | 1.729 (1.215–2.461) | 3.285 (2.128–5.072) | 1.733 (1.440–2.087) | 1.686 (1.380–2.059) |
| Temperature in working area | 0.973 (0.961–0.985) | 0.975 (0.960–0.989) | 0.964 (0.943–0.985) | 0.881 (0.846–0.918) | 0.992 (0.976–1.008) | 0.990 (0.973–1.006) |
| Ventilation system | 0.388 (0.299–0.503) | 0.757 (0.563–1.018) | 0.711 (0.478–1.058) | 1.076 (0.619–1.869) | 0.299 (0.208–0.432) | 0.541 (0.368–0.796) |
| Fixed breaks shifted in time | 1 | 1 | 1 | 1 | 1 | 1 |
| Fixed breaks not shifted in time | 0.266 (0.213–0.331) | 0.220 (0.174–0.278) | 0.051 (0.024–0.111) | 0.014 (0.006–0.032) | 0.385 (0.306–0.486) | 0.377 (0.296–0.480) |
| Breaks freely taken | 0.199 (0.114–0.348) | 0.372 (0.197–0.703) | 0.196 (0.109–0.352) | 0.666 (0.283–1.570) | - | - |
| Regular workers | 1 | 1 | - | - | - | - |
| Temporary and contract workers | 1.707 (1.415–2.060) | 0.994 (0.807–1.225) | - | - | - | - |

Abbreviations: OR, odds ratio; AOR, adjusted odds ratio; 95% CI, 95% confidence interval.

from the final analysis (S2 Table) hardly changed the results (AOR OAF per employee 0.983, 95% CI 0.969–0.997; AOR temperature 0.933, 95% CI 0.837–1. 039; interaction term 1.001, 95% CI 1.000–1.003; n = 1,702.).

## Discussion

### Key results

The aim of the study was to identify risk factors for COVID-19 among meat industry workers. Participating companies provided information on 19,072 employees in the twelve predefined

**Table 6. Results of the logistic regression analysis for COVID-19 infections in the sample with information on OAF.**

| Characteristics | Overall N = 2,786 | | Without delivery, anesthesia/slinging/hanging, slaughter n = 2,334 | | With interaction term n = 2,334 | |
|---|---|---|---|---|---|---|
| | OR (95% CI) | AOR (95% CI) | OR (95% CI) | AOR (95% CI) | OR (95% CI) | AOR (95% CI) |
| Minimum distance at least 1.5 meters | 1 | 1 | 1 | 1 | 1 | 1 |
| Minimum distance less than 1.5 meters | 2.555 (2.101–3.108) | 3.340 (2.711–4.117) | 2.617 (2.114–3.240) | 3.723 (2.929–4.734) | 2.617 (2.114–3.240) | 3.606 (2.828–4.598) |
| Temperature in working area | 0.946 (0.930–0.962) | 0.939 (0.921–0.959) | 0.958 (0.935–0.982) | 0.987 (0.952–1.024) | 0.742 (0.673–0.817) | 0.901 (0.821–0.990) |
| Maximal outdoor air flow (OAF) per employee | 1.000 (1.000–1.000) | 1.000 (1.000–1.000) | 0.994 (0.991–0.996) | 0.996 (0.993–0.999) | 0.959 (0.945–0.973) | 0.984 (0.971–0.996) |
| Interaction term temperature and OAF | - | - | - | - | 1.003 (1.002–1.005) | 1.001 (1.000–1.003) |
| Fixed breaks shifted in time | 1 | 1 | 1 | 1 | 1 | 1 |
| Fixed breaks not shifted in time | 0.345 (0.252–0.473) | 0.178 (0.128–0.247) | 0.305 (0.222–0.419) | 0.229 (0.152–0.345) | 0.305 (0.222–0.419) | 0.250 (0.165–0.379) |
| Break freely | - | - | - | - | - | - |
| Regular workers | 1 | 1 | 1 | 1 | 1 | 1 |
| Temporary and contract workers | 1.458 (1.153–1.842) | 1.375 (1.067–1.772) | 1.610 (1.264–2.051) | 2.047 (1.457–2.877) | 1.610 (1.264–2.051) | 1,862 (1.317–2.632) |

Abbreviations: OR, odds ratio; AOR, adjusted odds ratio; 95% CI, 95% confidence interval.

work areas plus others, if present, to look at the association between COVID-19 and workplace conditions. The study shows that workers in areas with lower temperature have higher chances of contracting COVID-19; workers with higher OAF per employee have lower chances. These associations persist even after the introduction of an interaction term and are further reinforced by this. Maintaining a distance of at least 1.5 m is only possible in some work areas (approx. 40%). The dismantling and packaging work areas seem to neglect this important safety measure frequently. However, an association between social distance not being maintained and the chance of infection was observed across all work areas.

Facilities provided information about specified interventions and prevention efforts that were implemented. Overall, the results show that all facilities had implemented interventions to reduce transmission or were preventing ongoing exposure within the workplace, including a SARS-CoV-2 testing strategy. The impact of the interventions is complicated to evaluate because of the heterogeneity in defining a 'time zero' for intervention onsets given the varying timepoints of answering the questionnaire.

## Strengths

One strength of the present study is that all plants with a known superspreading event were contacted by the BGN. Although working conditions were self-reported, we were able to visit two plants on-site. Both visits mostly confirmed the answers given by the companies and our assumption about the actual number of potentially exposed subjects during a shift. In both plants, the collection of the room temperature confirms the information from the questionnaire. Additional information will soon be available for humidity and carbon dioxide concentration in the work areas.

The study design allowed for an investigation of the relationships between COVID-19 cases and the working conditions through cross-sectional data for different plants. In addition, the effect estimates were adjusted for important confounders, so that the observed associations of OAF per employee, temperature and social distance with COVID-19 infections are largely unbiased.

## Limitations

Some limitations must be considered in our study: While creating the questionnaire, it was a challenge to summarize the operational conditions at the workplace so that these parameters could be solicited in a standardized manner. In addition to the existing heterogeneity of the work areas of slaughterhouses and meat processing plants, as well as businesses for meat and sausage production, delimiting the work areas is not always clear. For example, there was a company in which processing meat was carried out together with the packaging of the meat in one hall.

Companies could only provide information on COVID-19 cases where employees reported their company on a voluntary basis that they were positive. These numbers were mostly in line with reported figures from the press or figures sent to the BGN by the health authorities.

In addition, the complexity of different ventilation and cooling systems in the meat industry needed to be addressed with rather general questions. Therefore, the technical parameters considered here are not enough to describe the cooling and air conditioning systems. In many cases, the interrogated companies rely on the manufacturer's information for these parameters, which must be inquired about in detail and which mostly relate to maximum performance values. Depending on performance levels and production requirements, these maximum values can deviate from the actual values within the respective work areas.

In addition, the meat processing industry was aware of the dangers and the associated damage (financial and ideological) of a COVID-19 outbreak. Each company tried to remove or deactivate potential viruses from the air using the SARS-CoV-2 occupational safety standards and additional measures (filtration using HEPA filters or UVC radiation). Companies affected by an outbreak were forced to establish additional measures and hygiene concepts in order to restart production. In addition, the type of cooling, and the presence of air circulation and additional air purification present before the outbreaks could bias the results towards the null. In a next step, we will analyze the data with genuine information on the temporal effects of the relationship between the outdoor air flow, additional air purification and the risk of infection.

## Interpretation

The prevalence of COVID-19 in the participating companies with the highest superspreading events in Germany was 12.1%. However, it must be borne in mind that the company with the largest superspreading event—described in Günther et al. [6]—is not included in the results.

Based on the available results, it is possible to confirm the assumptions in the literature and our hypotheses that the interaction of these factors can explain the globally observed COVID-19 outbreaks in the meat and poultry industry [5–7].

By far the most affected work area was the deboning and meat processing area, which is characterized by problematic workplace conditions, including low temperature, low air exchange rates, and lack of social distancing between workers.

The main analysis showed that employees working in an area with a ventilation system had a lower chance of becoming COVID-19. As suspected, it could be shown that very high levels of OAF per employee are achieved during slaughter, which biases the relationship between OAF per employee and COVID-19 towards the null. In the analysis with OAF, it was shown that with higher OAF per employee, employees had a smaller chance of becoming COVID-19 and an inverse relationship between temperature and infection.

The results for the different break times are unexpected, as one would assume that a temporal shift leads to smaller groups and thus a lower infection risk. The results may be explained by an unmeasured factor though, such as the canteen size and its ventilation.

Other factors that we were unable to consider for survey reasons are the wearing of face covering and the accommodation and the transportation mode to work of the temporary and contract workers. The extent to which face coverings were worn in the different work areas was collected in the questionnaire as well, but since a face covering was required almost everywhere after the pandemic began, we decided to exclude this variable from the model. Data were collected on accommodation and transportation to work of temporary and contract workers, but these were not available at the work area level and therefore not included in the main analysis.

In order to better assess our OAF values, we looked for ways to match them with official recommendations. According to the workplace ordinance for indoor spaces (ASR A3.6), depending on the activity of the employees or their level of activity, an indoor air quality considered beneficial is achieved if the concentration of exhaled carbon dioxide (Pettenkofer number) does not exceed a value of 1000 ppm. A carbon dioxide value of 1000 ppm corresponds to an outdoor air flow per person and per hour of $81m^3/h$ during heavy physical activity. The recurrent emergence of such outbreaks suggests that the Pettenkofer value should be kept as far below 1000 ppm as possible, which increases the necessary outdoor air flow per person [21]. In our analysis and in the two on-site visits, the problematic areas are mainly the deboning and meat cutting, meat production, and packaging areas, which did not fulfill the recommended outdoor air flow.

### Generalizability

In conclusion, this study indicates that there are multiple risk factors for the transmission of SARS-CoV-2 in the work environments of meat and poultry processing companies. The different ventilation and temperature conditions suggest that each outbreak may develop differently. Both, the different courses of the outbreaks and the heterogeneity of the company's specialization and structural conditions make it difficult to make general statements about the infection occurrence in the meat industry. However, our results illustrate the benefit of preventive measures and the need to further investigate the ventilation conditions on-site. In any case, the amount of OAF per hour per person should be determined as the target value for air exchange. This seems even more important when considering the interaction with temperature.

To what extent the results can be transferred to private living spaces, offices or classrooms seems questionable. These settings work with intensive ventilation for a short time period for a virus dilution in the air of the rooms, contrary to the permanent ventilation in the meat industry.

The significance of this study is imminent for the meat and poultry processing industry but should also play a role in other food processing industries, distribution centers and other workplaces where employees work under similar conditions [2,22]. It points to the importance of air quality and air flow in confined spaces as a valuable additional measure to prevent future superspreading events.

## Supporting information

**S1 Table. COVID-19 among workers in meat and poultry processing plants by work area.** (DOCX)

**S2 Table. Results of the logistic regression analysis COVID-19 infections in the sub-analysis.** Abbreviations: OR, odds ratio; AOR, adjusted odds ratio; 95% CI, 95% confidence interval; OAF outdoor air flow.
(DOCX)

**S1 Appendix. Public Health Clinic-Patient Satisfaction Questionnaire (PHC-PSQ).** (PDF)

**S2 Appendix. Public Health Clinic-Patient Satisfaction Questionnaire (PHC-PSQ).** (PDF)

**S3 Appendix. Public Health Clinic-Patient Satisfaction Questionnaire (PHC-PSQ).** (PDF)

## Acknowledgments

This collection of information and analysis would not have been possible without the willingness and interest of the companies and a trusting cooperation with the BGN; the study contributes to a further expansion of this good dialogue process. Therefore, we want to thank all participating plants. We would also like to thank Prof. Thomas Behrens (IPA), who read the manuscript critically and supported us with helpful insights and Ludger Stennes (BGN), who collected the data and helped with insights to the meat processing.

## Author Contributions

**Conceptualization:** Roman Pokora, Susan Kutschbach, Matthias Weigl, Detlef Braun, Annegret Epple, Eva Lorenz, Stefan Grund, Juergen Hecht, Helmut Hollich, Peter Rietschel, Frank Schneider, Roland Sohmen, Isabel Dienstbuehl.

**Data curation:** Susan Kutschbach, Matthias Weigl, Annegret Epple, Peter Rietschel, Frank Schneider.

**Formal analysis:** Roman Pokora, Susan Kutschbach, Annegret Epple, Eva Lorenz, Peter Rietschel, Frank Schneider, Katherine Taylor.

**Investigation:** Roman Pokora, Matthias Weigl, Annegret Epple, Peter Rietschel.

**Methodology:** Roman Pokora, Susan Kutschbach, Matthias Weigl, Detlef Braun, Eva Lorenz, Stefan Grund, Juergen Hecht, Helmut Hollich, Frank Schneider, Roland Sohmen.

**Project administration:** Roman Pokora, Detlef Braun, Stefan Grund, Juergen Hecht, Helmut Hollich, Peter Rietschel, Frank Schneider, Roland Sohmen, Isabel Dienstbuehl.

**Resources:** Roman Pokora.

**Supervision:** Matthias Weigl, Annegret Epple, Isabel Dienstbuehl.

**Validation:** Roman Pokora, Susan Kutschbach, Matthias Weigl, Annegret Epple.

**Writing – original draft:** Roman Pokora, Susan Kutschbach, Matthias Weigl, Annegret Epple, Eva Lorenz, Katherine Taylor.

**Writing – review & editing:** Detlef Braun, Stefan Grund, Juergen Hecht, Helmut Hollich, Peter Rietschel, Frank Schneider, Roland Sohmen, Isabel Dienstbuehl.

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
