## [Decision Letter · Decision Letter 0]

9 Feb 2021

PONE-D-20-37878

Investigation of superspreading COVID-19 outbreaks events in meat and poultry processing plants in Germany: A cross-sectional study

PLOS ONE

Dear Dr. Pokora,

Thank you for submitting your manuscript to PLOS ONE. After careful consideration, we feel that it has merit but does not fully meet PLOS ONE’s publication criteria as it currently stands. Therefore, we invite you to submit a revised version of the manuscript that addresses the points raised during the review process.

Your paper has been scientifically assessed, and the comments provided by our reviewer provide good account of the good quality of the work you submitted, However, the reviewer asks for some improvements and clarifications (most of them relatively minor) before considering suggesting the acceptance of the paper in PLOS ONE. Please ensure to respond all them with enough level of detail and seiousness during your revisions. Therefore, I will be able to make an editorial decision as soon as I receive the second reviewer's report.

We look forward to receiving your revised manuscript.

Kind regards,

Sergio A. Useche, Ph.D.

Academic Editor

PLOS ONE

Journal Requirements:

2) Please include additional information regarding the survey or questionnaire used in the study and ensure that you have provided sufficient details that others could replicate the analyses. For instance, if you developed a questionnaire as part of this study and it is not under a copyright more restrictive than CC-BY, please include a copy, in both the original language and English, as Supporting Information, or include a citation if it has been published previously.

3) In the Methods, please discuss whether and how the questionnaire was validated and/or pre-tested. If these did not occur, please provide the rationale for not doing so.

4) Please provide further details on sample size and power calculations.

5) In statistical methods, please refer to any post-hoc corrections to correct for multiple comparisons during your statistical analyses. If these were not performed please justify the reasons. Please refer to our statistical reporting guidelines for assistance (https://journals.plos.org/plosone/s/submission-guidelines.#loc-statistical-reporting).

6) In your statistical analyses, please state whether you accounted for clustering by locality, i.e, processing plant. For example, did you consider using multilevel models?

7) In your discussions and conclusions please take care to avoid statements implying causality from correlational research. For example, avoid the use of terms such as “effect of” or “resulted in." Instead consistently use terms such as "associated with" or "associations."

8) We note that you have indicated that data from this study are available upon request. PLOS only allows data to be available upon request if there are legal or ethical restrictions on sharing data publicly. For information on unacceptable data access restrictions, please see http://journals.plos.org/plosone/s/data-availability#loc-unacceptable-data-access-restrictions.

Reviewers' comments:

Reviewer's Responses to Questions

**Comments to the Author**

1. Is the manuscript technically sound, and do the data support the conclusions?

Reviewer #1: Yes

2. Has the statistical analysis been performed appropriately and rigorously? 

Reviewer #1: Yes

3. Have the authors made all data underlying the findings in their manuscript fully available?

Reviewer #1: Yes

4. Is the manuscript presented in an intelligible fashion and written in standard English?

Reviewer #1: Yes

5. Review Comments to the Author

Reviewer #1: The authors have conducted a cross-sectional analysis of meat processing plants to assess factors associated with risk for transmission of COVID-19. They surveyed 26 plants and included 7 companies with more than 10 cases in their analyses. The results identified lack of social distancing, cool temperature and lack of air exchange as factors associated with higher rates of transmission. These findings are consistent with other reports and provide companies with guidance for reducing the risk of transmission in their facilities. While useful, there are several issues that need to be addressed.

The survey identified plants with a range of infection rates. There was no comparison to identify differences between plants with high infection rates versus low infection rates. It seems that this could have been useful, and there should be some comment about why this wasn’t pursued, or if it was pursued, why it wasn’t reported.

Line 61-64. It is not clear how contract workers mentioned here relate to the temporary or contract workers who comprised the main study population. It is somewhat confusing throughout to understand the differences between permanent, contract and temporary workers. Please elaborate.

Lines 110-111 The distribution of companies by cases is based on the interval June-September?

Line 112 What is “paper and pencil questionnaire”? Is this just to distinguish from an online survey?

Line 142 “point prevalence” seems really to be more of a period-based incidence measure.

Line 147 Why were there no comparisons between plants with different categories of infection?

Line 150 Is the difference between fixed breaks with or with time shifts a reflection on whether every takes a break at the same time or not? This could be clarified.

Line 162 The calculation assumes that all shifts have the same number of employees. Is that correct?

Lines 191-194 Are these calculations manipulations of the model to predict outcomes for defined exposure levels? This warrants a bit more description.

Line 210 What does it mean that one plant took part although it was not contacted?

Table 1 Why were employees excluded? In Table 1 there is a column heading for contract workers. In Table 3 the corresponding column headings mentions regular workers. These appear to be the same and should be called the same.

6. PLOS authors have the option to publish the peer review history of their article (what does this mean?). If published, this will include your full peer review and any attached files.

Reviewer #1: No

---

## [Author Response · Author response to Decision Letter 0]

16 Apr 2021

Comments of the editor and our replies:

Thank you. We revised the headings and tables.

2) Please include additional information regarding the survey or questionnaire used in the study and ensure that you have provided sufficient details that others could replicate the analyses. For instance, if you developed a questionnaire as part of this study and it is not under a copyright more restrictive than CC-BY, please include a copy, in both the original language and English, as Supporting Information, or include a citation if it has been published previously.

We have added both the original questionnaire in German in the first and final version and a translated version of the final questionnaire as supporting information. 

3) In the Methods, please discuss whether and how the questionnaire was validated and/or pre-tested. If these did not occur, please provide the rationale for not doing so.

After analyzing the answers for the questionnaire 1, we discovered that we were not able to distinguish between regular, contract and temporary workers and added therefore a new column in a revised version of the questionnaire. Additionally, we adapted the questions about the air conditioning. In version 1 we just collected the proportion of outdoor air and air circulation. After carefully reading the literature we decided that it would be more appropriate to collect information on outdoor air flow and to use this information. After that decision we contacted again the organizations and collected this additional information. We included this in the manuscript in line 135ff.

4) Please provide further details on sample size and power calculations.

The first version of the questionnaire was developed to collect information on the outbreaks to better support the organizations we insure. In the second step, we considered whether the data could be analyzed combined. Until this point in time we knew about three of four outbreak events in meat and poultry plants in Germany and could not assess the future extent. Therefore, we did not calculate a sample size calculation. At the time point of the analysis the sample size was sufficiently large, that no formal power analysis was carried out.

5) In statistical methods, please refer to any post-hoc corrections to correct for multiple comparisons during your statistical analyses. If these were not performed please justify the reasons. Please refer to our statistical reporting guidelines for assistance.

We understand the concerns from PLOS ONE and agree with the request. For that reason, we do not report P values anywhere in the manuscript. The reasons for doing so:

1. The sample size is sufficiently large that even seemingly trivial differences are likely to be statistically significant. 

2. P-values are commonly misinterpreted as the probability that the test hypothesis is true, or as the probability that observed association is due to chance alone. Both are false.

In sum, in the present study we do not believe that adding P values or corrected P values would help readers to get a better sense of whether or not the reported characteristics differ across working groups, or that P values and significance testing would add valuable information beyond what is already reported in the manuscript.

We hope that the explanations will satisfy you and that you will appreciate our efforts to conform with the STROBE statement and the position of the American Statistical Association (2016).

6) In your statistical analyses, please state whether you accounted for clustering by locality, i.e, processing plant. For example, did you consider using multilevel models?

That is a very interesting point and we considered using multilevel models. In our understanding, hierarchical data in our study would be individual data of the workers, such as nationality or the disease status, (subordinate level 1) in the different departments or companies (parent level 2). In our evaluation, we have 2 levels, where level 2 is the parent level and level 1 is the child level. The predictors (UVs) can be at different levels and can be used for predictions. The dependent variable, on the other hand, is always at level 1. Since all the predictors are at level 2 in our study, we decided to evaluate a classical statistical method.

7) In your discussions and conclusions please take care to avoid statements implying causality from correlational research. For example, avoid the use of terms such as “effect of” or “resulted in." Instead consistently use terms such as "associated with" or "associations."

Thank you! It was not in our interest to create this impression and we carefully went through the manuscript again and changed the wording.

8) We note that you have indicated that data from this study are available upon request. PLOS only allows data to be available upon request if there are legal or ethical restrictions on sharing data publicly. For information on unacceptable data access restrictions, please see http://journals.plos.org/plosone/s/data-availability#loc-unacceptable-data-access-restrictions.

Thank you. We made the dataset for the main analysis available and changed our statement:

Availability of data and material: All relevant data are within the manuscript and the dataset is available under https://doi.org/10.5281/zenodo.4692642. The legal basis for the collection of the study data is section 1, section 9 paragraph 8, section 14 and 207 of the Social Code (SGB VII) and the statutes of the BGN. All persons involved in the project are subject to the obligation to maintain confidentiality in accordance with Section 35 SGB I. The analysis presents insurance data collected during the legal mandate with ongoing follow-up data collection. Original questionnaires and measurement reports are not made available for the scientific community outside the established and controlled workflows and algorithms. To meet the general idea of verification and reproducibility of scientific findings, we offer access to the original data at the local database in accordance with the data protection protocol upon request at any time. The study team, which constitutes a member of each involved department and the coordinating principal investigator of the study (Dr Pokora) decides on internal and external access of researchers and use of the data based on a research proposal to be supplied by the researcher. Interested researchers make their requests to the coordinating principal investigator of the study (roman.pokora@bgn.de).

Comments of the reviewer and our replies:

Comments to the Author

1. Is the manuscript technically sound, and do the data support the conclusions?

Reviewer #1: Yes

2. Has the statistical analysis been performed appropriately and rigorously? 

Reviewer #1: Yes

3. Have the authors made all data underlying the findings in their manuscript fully available?

Reviewer #1: Yes

4. Is the manuscript presented in an intelligible fashion and written in standard English?

Reviewer #1: Yes

5. Review Comments to the Author

Reviewer #1: The authors have conducted a cross-sectional analysis of meat processing plants to assess factors associated with risk for transmission of COVID-19. They surveyed 26 plants and included 7 companies with more than 10 cases in their analyses. The results identified lack of social distancing, cool temperature and lack of air exchange as factors associated with higher rates of transmission. These findings are consistent with other reports and provide companies with guidance for reducing the risk of transmission in their facilities. While useful, there are several issues that need to be addressed.

The survey identified plants with a range of infection rates. There was no comparison to identify differences between plants with high infection rates versus low infection rates. It seems that this could have been useful, and there should be some comment about why this wasn’t pursued, or if it was pursued, why it wasn’t reported.

Line 61-64. It is not clear how contract workers mentioned here relate to the temporary or contract workers who comprised the main study population. It is somewhat confusing throughout to understand the differences between permanent, contract and temporary workers. Please elaborate.

We have clarified our wording and the problems with the wording. After the evaluation of the first version of the questionnaire we decided to combine contract and temporary workers. The definition and the exact difference of the two groups of temporary and contract workers is not selective and we tried to clarify our use in the method section and added a section in the method part in line 141ff.

Lines 110-111 The distribution of companies by cases is based on the interval June-September?

Yes, data was collected between June and September 2020. Earliest outbreak by a participant plant was May 2020 and was the first outbreak in a meat processing facility in Germany. 

Line 112 What is “paper and pencil questionnaire”? Is this just to distinguish from an online survey?

Since the companies were given a word document to fill out, we changed this word to “questionnaire”.

Line 142 “point prevalence” seems really to be more of a period-based incidence measure.

We agree with the reviewer that the wording is here a little bit problematic. The best description is maybe to describe each outbreak as a “point prevalence” and the combined prevalence from all companies a “period-based prevalence”. We clarified that in the manuscript.

Line 147 Why were there no comparisons between plants with different categories of infection?

We analyzed the differences across plants with different categories of infection. The comparison is unfortunately a little bit problematic because we had to analyze free text and put it in categories. Also, the occupational safety measures display safety measures at different time points. Now we included some intervention and prevention efforts of the facilities (line 234ff) and put a new section in the method part as well in the discussion section.

Line 150 Is the difference between fixed breaks with or with time shifts a reflection on whether every takes a break at the same time or not? This could be clarified.

Yes. We clarified that and added an explanation: “Fixed breaks without time shift mean that all workers have their breaks at the same time. Fixed breaks with time shift mean that there is a break pattern where different working groups, departments or divisions have different time windows for their break.”

Line 162 The calculation assumes that all shifts have the same number of employees. Is that correct?

Yes, the assumption is that each shift in a company has the same number of employees. Like we wrote in the discussion the visits in the companies confirmed our assumption about the actual number of potentially exposed subjects during a shift.

Lines 191-194 Are these calculations manipulations of the model to predict outcomes for defined exposure levels? This warrants a bit more description.

Exactly. These calculations are based on the same model and only the amount of fresh air has been changed. We excluded this from the manuscript, because it does not add any further knowledge to the association of the variables.

Line 210 What does it mean that one plant took part although it was not contacted?

It means that the parent company has distributed the questionnaire to all of its companies and all companies of the parent company have answered the questionnaire. Therefore, we received one questionnaire from a company we did not write to.

Table 1 Why were employees excluded? 

Employees were excluded when information was missing. We discussed the possibility of imputation, but because our analysis was only possible because working areas in the meat processing area are clearly distinct, we decided that we don’t want to impute missing values, for example for the cleaning personal. Our procedure is explained at the end of the statistical analysis part. 

In Table 1 there is a column heading for contract workers. In Table 3 the corresponding column headings mentions regular workers. These appear to be the same and should be called the same.

We agree and clarified our wording into regular, contract and temporary workers. Thank you for this advice.

---

## [Decision Letter · Decision Letter 1]

24 May 2021

Investigation of superspreading COVID-19 outbreak events in meat and poultry processing plants in Germany: A cross-sectional study

PONE-D-20-37878R1

Dear Dr. Pokora,

We’re pleased to inform you that your manuscript has been judged scientifically suitable for publication and will be formally accepted for publication once it meets all outstanding technical requirements.

Kind regards,

Sergio A. Useche, Ph.D.

Academic Editor

PLOS ONE

Additional Editor Comments (optional):

Reviewers' comments:

Reviewer's Responses to Questions

**Comments to the Author**

1. If the authors have adequately addressed your comments raised in a previous round of review and you feel that this manuscript is now acceptable for publication, you may indicate that here to bypass the “Comments to the Author” section, enter your conflict of interest statement in the “Confidential to Editor” section, and submit your "Accept" recommendation.

Reviewer #1: All comments have been addressed

2. Is the manuscript technically sound, and do the data support the conclusions?

Reviewer #1: Yes

3. Has the statistical analysis been performed appropriately and rigorously? 

Reviewer #1: Yes

4. Have the authors made all data underlying the findings in their manuscript fully available?

Reviewer #1: Yes

5. Is the manuscript presented in an intelligible fashion and written in standard English?

Reviewer #1: Yes

6. Review Comments to the Author

Reviewer #1: The authors have addressed all relevant comments. The manuscript can now be accepted for publication.

7. PLOS authors have the option to publish the peer review history of their article (what does this mean?). If published, this will include your full peer review and any attached files.

Reviewer #1: No

---

## [Editor Report · Acceptance letter]

28 May 2021

PONE-D-20-37878R1 

Investigation of superspreading COVID-19 outbreak events in meat and poultry processing plants in Germany: A cross-sectional study 

Dear Dr. Pokora:

I'm pleased to inform you that your manuscript has been deemed suitable for publication in PLOS ONE. Congratulations! Your manuscript is now with our production department. 

Kind regards, 

on behalf of

Dr. Sergio A. Useche 

Academic Editor

PLOS ONE